# Sugar- and Artificially Sweetened Beverages Consumption Linked to Type 2 Diabetes, Cardiovascular Diseases, and All-Cause Mortality: A Systematic Review and Dose-Response Meta-Analysis of Prospective Cohort Studies

**DOI:** 10.3390/nu13082636

**Published:** 2021-07-30

**Authors:** Yantong Meng, Siqi Li, Jabir Khan, Zijian Dai, Chang Li, Xiaosong Hu, Qun Shen, Yong Xue

**Affiliations:** 1National Engineering and Technology Research Center for Fruits and Vegetables, College of Food Science and Nutritional Engineering, China Agricultural University, Beijing 100083, China; mengyantong@cau.edu.cn (Y.M.); lisiqi@cau.edu.cn (S.L.); LS20193060048@cau.edu.cn (J.K.); S20193060887@cau.edu.cn (Z.D.); lichang061@sina.com (C.L.); huxiaos@263.net (X.H.); 2Key Laboratory of Plant Protein and Grain Processing, College of Food Science and Nutritional Engineering, China Agricultural University, Beijing 100083, China; 3Xinghua Industrial Research Centre for Food Science and Human Health, China Agricultural University, Xinghua 225700, China

**Keywords:** sugar-sweetened beverages, artificially sweetened beverages, type 2 diabetes, cardiovascular diseases, all-cause mortality

## Abstract

Although studies have examined the association between habitual consumption of sugar- (SSBs) and artificially sweetened beverages (ASBs) and health outcomes, the results are inconclusive. Here, we conducted a dose-response meta-analysis of prospective cohort studies in order to summarize the relationship between SSBs and ASBs consumption and risk of type 2 diabetes (T2D), cardiovascular diseases (CVDs), and all-cause mortality. All relevant articles were systematically searched in PubMed, Embase, and Ovid databases until 20 June 2020. Thirty-four studies met the inclusion criteria and were eligible for analysis. Summary relative risks (RRs) and 95% confidence intervals (95% CI) were estimated using random effects or fixed-effects model for highest versus lowest intake categories, as well as for linear and non-linear relationships. With each additional SSB and ASB serving per day, the risk increased by 27% (RR: 1.27, 95%CI: 1.15–1.41, *I*^2^ = 80.8%) and 13% (95%CI: 1.03–1.25, *I*^2^ = 78.7%) for T2D, 9% (RR: 1.09, 95%CI: 1.07–1.12, *I*^2^ = 42.7%) and 8% (RR: 1.08, 95%CI: 1.04–1.11, *I*^2^ = 45.5%) for CVDs, and 10% (RR: 1.10, 95%CI: 0.97–1.26, *I*^2^ = 86.3%) and 7% (RR: 1.07, 95%CI: 0.91–1.25, *I*^2^ = 76.9%) for all-cause mortality. Linear relationships were found for SSBs with T2D and CVDs. Non-linear relationships were found for ASBs with T2D, CVDs, and all-cause mortality and for SSBs with all-cause mortality. The findings from the current meta-analysis indicate that increased consumption of SSBs and ASBs is associated with the risk of T2D, CVDs, and all-cause mortality.

## 1. Introduction

Sugar-sweetened beverages (SSBs, also categorized as sugary drinks or soft drinks) refer to the beverages with added sugar or sweeteners that have been prevalent worldwide, especially among younger adults, with their intake frequency and portion size increasing markedly in most countries over the last several decades [1,2,3,4,5,6]. During 2011–2014, about one-half of US adults consumed at least one SSB per day, which contributed 6.9% of total energy intake for men and 6.1% for women [7] and over 46% of added sugar in the US diet [8]. In Asia, the energy contribution of SSBs among adults increased from 32 to 82 kcal per day in South Korea during 1998–2009 [9]. Data from Chinese Nutrition and Health Surveillance 2010–2012 showed that half of Chinese adults consumed SSBs, and the prevalence in cities was similar to it in counties [10]. Globally, although many governments have initiated actions to reduce the consumption of SSBs in the last few years, however, the intake of SSBs has remained fairly stable [11]. Artificially sweetened beverages (ASBs), referring to non-nutritive sweetened beverages, are often suggested as alternatives to SSBs for those who want to reduce sugars and calories, making it logical that artificial sweeteners would have provided less weight gain and prevented cardio-metabolic risks [12]. In this regard, the number of adults who consume ASBs has substantially risen in the US [13], and purchases of carbonated ASBs increased by 24% during 1997–2011 in Australia [14]. However, it remains unclear whether the replacement of sugar-sweetened products with those containing artificial sweeteners is actually beneficial.

Type 2 diabetes (T2D), characterized by increased blood glucose levels and insulin resistance, is a major risk factor for cardiovascular diseases (CVDs), which is mainly responsible for half the mortality in T2D patients [15]. As a major source of added sugar in a diet, there are links between the intake of SSBs and risks of obesity [16], diabetes [17], coronary heart disease [18], stroke [19], and mortality [20] that were evaluated in epidemiological studies. Previous evidence from large observational studies demonstrated inconsistent results on the links between habitual SSBs consumption and development of diabetes, CVDs, and mortality, although general studies revealed modest relationships [16,17,18,19,20,21,22]. For example, the results from a prospective, community-based cohort in the US showed that SSBs consumption was not consistent with the incidence of T2D [21]. Similarly, no association between SSBs consumption and mortality was found in the Singapore Chinese Health Study [22]. Although six artificial sweeteners (including acesulfame potassium, aspartame, saccharin, sucralose, neotame, and stevia) have been considered safe for human consumption by the US Food and Drug Administration (FDA), the associations of ASBs consumption and risks of the above-mentioned diseases have been controversial for a long time [20,23]. For example, a prospective cohort of 81,714 postmenopausal US women found a strong correlation between ASBs consumption and all-cause mortality after 11.9 years [24]. In contrast, a prospective cohort study of 37,716 men found that the consumption of SSBs, but not ASBs, was associated with a higher risk of mortality [20]. In the meta-analysis studies that have been conducted [3,25,26], the aspects concerned are not comprehensive enough, the analyses are not sufficient, and some new studies have been published [27,28]. Furthermore, people usually pay less attention to the relationship between ASBs and health outcomes.

Thus, we performed an updated comprehensive systemic review and a meta-analysis of all relevant prospective cohort studies published in recent years to synthesize knowledge of the association between SSBs and ASBs consumption and risks of T2D, CVDs, and all-cause mortality and to explore their dose-response relationships. The aim of this article is to provide health evidence for the relationship between beverages and health outcomes based on previous research and to give people a clear understanding of SSBs and ASBs.

## 2. Materials and Methods

### 2.1. Search Strategy

This report has been prepared in accordance with the PRISMA guidelines [29]. We systematically searched the relevant studies in three databases until 20 June 2020: PubMed, Embase, and Ovid. Search terms included words related to types of beverages, diabetes, cardiovascular diseases, all-cause mortality, and prospective study design (Appendix A). We used “sugar-sweetened beverages”, “soft drinks”, “diabetes”, “CVD”, “all-cause mortality”, “prospective”, and other words as the keywords for the systematic search [25]. Additionally, the reference lists of searched articles were reviewed to identify any studies that were not selected from the preliminary searches. The language of publications in our systematic search was limited to English papers, and no time limitations were applied.

### 2.2. Eligibility and Study Selection

After the removal of duplications, two authors (Y.M and S.L) independently reviewed the titles and abstracts of all studies and selected eligible studies based on the following criteria: (1) prospective design (cohort, case-cohort, and nested case-control); (2) measured and reported SSBs or ASBs as exposure and in at least two categories, and reported T2D, CVDs, or all-cause mortality as the outcomes; (3) a study population that was healthy at baseline; (4) studies that reported relative risks (RRs), hazard ratios (HRs), or odds ratios (ORs) with 95% confidence intervals (CIs), or data to calculate these; (5) the language is English. We excluded repetitive articles, letters, comments, reviews, meta-analyses, studies with incomplete, incorrect or uncertain data, and papers with unavailable full texts. Any disagreements between the two authors were resolved through discussion under the supervision of a third author (Y.X).

### 2.3. Data Extraction and Quality Assessment

According to our search strategy, 11,822 published articles were identified. After removing 4982 duplicates, 6840 articles were reserved for further review; 6806 articles were excluded after screening for titles and abstracts. Finally, 34 relevant articles were selected for the systematic review and meta-analysis on the basis of full-text reviews.(Figure 1). For the selected eligible studies, two authors (Y.M and S.L) independently reviewed the full texts and extracted the following data: first author’s last name, geographical region where the study was conducted, name of the study, follow-up duration, total number of individuals, mean age and/or age range at baseline, gender, beverage consumption (method of assessment, the highest category versus the lowest category), disease (method of assessment, number of cases), confounding factors that were adjusted in the analysis, and RRs, HRs, or ORs estimates with corresponding 95% CIs for each category. If an included study reported several adjustment models of risk estimates, we extracted the fully adjusted effect sizes. The quality of every selected eligible article was assessed on the basis of the 9-point Newcastle–Ottawa quality assessment scale for observational studies, including three aspects: selection of the study groups (four items, one point each), comparability of the groups (one item, up to two points), and ascertainment of the outcome of interest (three items, one point each) in eight questions. The total score is calculated by summing up the score for each answer, and the articles of more than six scores were considered to be of high quality. Any disagreements between the two reviewers (Y.M and S.L) were resolved through discussion under the supervision of the third researcher (Y.X).

### 2.4. Statistical Methods

All statistical analyses were performed using STATA version 15.1 (Stata Corp., College Station, TX, USA). The intake of SSBs or ASBs was considered as the main exposure. We defined SSBs as any sweetened beverages, including sweet sugar drinks, sweetened cola, sugar-sweetened soft drinks, sugar-sweetened sodas, orange juices, apple juices, and other fruit juices, but did not include diet or non-caloric beverages. The ASBs referred to low caloric soft drinks as reported in each study. The RRs and 95% CIs were considered as the effect size, and the reported HRs, or ORs were considered to be equal to RRs. We pooled the reported risk estimates of the highest level compared to the lowest category of SSBs or ASBs [30]. The heterogeneity between studies was tested by using Cochrane’s Q test of heterogeneity and the *I*^2^ statistic [31]. *I*^2^ greater than 50% was considered as significant heterogeneity among studies, and random-effects models could account for the variation, which provided more conservative results than fixed-effects models [32]. On the contrary, *I*^2^ lower than 50% were assessed as non-significant heterogeneity among studies and conducted fixed-effects models for meta-analysis. We performed several subgroup analyses based on gender, follow-up duration (≤10 or >10 years), geographical region (North America, Europe, or Asia), number of participants (≤10,000, or >10,000), adjustments for main confounding variables (BMI, alcohol consumption, smoking status, physical activity, education, energy intake, fruits and vegetables, processed meat, hypertension, and diabetes at baseline) to test the association between each factor and each result and to find the potential source of heterogeneity. Sensitivity analyses were evaluated to explore that whether the pooled RR could be significantly affected by a single study. The potential publication biases were evaluated by using funnel plots and further tested by Begg’s test [33] and Egger’s test [34]. *p*-values < 0.05 were considered as statistically significant for all tests.

Additional dose-response analyses were performed by using all available data points from each study. The studies with reported sufficient information were eligible for inclusion if they: (1) considered the lowest category as the reference; (2) reported beverage intake in at least three categories; (3) reported the range, median, or mean of each category; (4) reported the number of cases and participants in each category of beverage intake; (5) reported adjusted RR, HR, or OR with 95% CI of each category. For the highest category without an upper limit, we used the lower limit plus 1/2 of the range of interval of the previous category to estimate [35]. When the numbers of cases and participants in each category were not reported, we divided the total number of cases and participants by the number of categories [36,37]. In the case of articles that we were unable to extract data for dose-response analyses [19,20,38,39,40,41,42,43], we contacted corresponding authors for the key information, but we received no reply. According to the methods developed by Greenland and colleagues, we measured the dose-response relationships using generalized least-squares trend estimation [44,45]. We calculated the RR for each additional SSB and ASB serving to estimate the dose-response relationship in each included study, and we examined the linear or non-linear dose-response relationships for all included studies. If a *p*-value conducted by the fixed-effects model was greater than 0.05, which meant that the null hypothesis could not be rejected, we used the fixed-effects model; on the contrary, the random-effects model was chosen. The regression parameter test was performed to determine the linear or non-linear model, and the linear model would be selected when the *p*-value was above 0.05.

## 3. Results

### 3.1. Beverage Intake and T2D

We identified 17 prospective cohort studies with 645,658 participants from nine countries and regions that were eligible for the analysis of beverage intake and the risk of T2D. Two studies included only males [17,46], six studies included only females [38,40,47,48,49,50], and nine studies included both sexes [21,39,51,52,53,54,55,56,57]. All participants were adults, with ages ranging from 18 to 79 years old, and the follow-up durations were ranging from 5.5 to 30 years. We extracted 20 groups of data from 17 studies about SSBs and nine groups of data from nine studies about ASBs. The consumption of beverages was recorded by different ways, such as FFQ [17,21,38,39,40,47,49,50,56,57], diet history [48,54], and alternative ways [46,51,52,53,55]. The maximum and minimum intakes in studies were extracted and compared. Methods of diagnosis and confirmed cases of T2D were recorded, which included self-reporting [17,38,40,47,51,57], medical diagnosis [39,46,53,54,56], or both of them [21,48,49,50,52,55]. The adjustments in each study were counted. Nearly all studies assessed age (with the exception of [48,52]), BMI (with the exception of [40,49,50]), physical activity (with the exception of [46]), and all studies assessed smoking status. Based on the Newcastle–Ottawa Scale, all studies were of a high quality (≥6 scores). Detail information is shown in Appendix A.

#### 3.1.1. SSBs and T2D

The consequence indicated that the highest category compared to the lowest category of SSBs intake was associated with a 29% higher risk of T2D in these cohorts (RR: 1.29; 95% CI: 1.23–1.34), with low evidence of heterogeneity (*I*^2^ = 29.9%, *P*_heterogeneity_ = 0.102) (Figure 2a). No particular study had a significant influence on the summary effect in a sensitivity analysis. We used the funnel plot to evaluate publication bias and found it was basically symmetrical (Figure 3a). Then we further used Begg’s test and Egger’s test to determine that there was no evidence of significant publication bias (Begg’s test: 0.496, Egger’s test: 0.446). 

A total of 13 studies were included to analyze the dose-response relationship [17,21,46,48,49,50,51,52,53,54,55,56,57]. With each additional serving of SSB per day, the risk of developing T2D increased by 27% (RR: 1.27, 95%CI: 1.15–1.41), with high evidence of heterogeneity (*I*^2^ = 80.8%, *P*_heterogeneity_ < 0.001) (Figure 4a). We sequentially removed each study from the pooled analysis, but the heterogeneity remained stable. The funnel plot (Figure 5a), the Begg’s test (*p* = 0.344), and the Egger’s test (*p* = 0.016) were used to evaluate publication bias. The results showed a linear relationship between T2D and SSBs intake (Figure 6a).

#### 3.1.2. ASBs and T2D

ASBs intake contributed to an 18% higher risk of T2D in these cohorts (RR: 1.18; 95% CI: 1.08–1.29) when compared the highest category with the lowest category, with moderate evidence of heterogeneity (*I*^2^ = 53.5%, *P*_heterogeneity_ = 0.028) (Figure 2b). One large prospective cohort study [38] accounted for all of the observed heterogeneity, and after this study was eliminated from the pooled analysis, the heterogeneity disappeared, and the association changed to 1.22 (95% CI: 1.11, 1.34; *I*^2^ = 31.1%, *P*_heterogeneity_ = 0.180). There were nine groups of data in ASBs intake and T2D, less than 10 groups. According to the Cochrane Handbook, we did not evaluate publication bias. 

Seven studies were included to analyze the dose-response relationship [17,46,47,48,52,54,55,58,59,60,61,62,63,64], the pooled RR for T2D for each additional ASB serving per day was 1.13 (95%CI: 1.03–1.25), with high evidence of heterogeneity (*I*^2^ = 78.7%, *p* < 0.001) (Figure 4b). After removing every study from the analysis, the heterogeneity remained stable. We did not evaluate publication bias because there were less than 10 groups. There was a non-linear relationship between T2D and ASBs intake according to our analysis (Figure 6b).

### 3.2. Beverage Intake and CVDs

Ten prospective cohort studies with 582,082 participants from three countries were eligible for the analysis of beverage intake and the risk of CVDs. There were three male studies [41,63,64], four female studies [24,41,60,61], and four studies for both sexes [19,58,59,62]. All participants were adults, with ages ranging from 30 to 79 years old, and the follow-up durations were ranging from 9.8 to 28 years. A total of 14 groups of data were extracted from nine studies about SSBs, and eight groups of data were extracted from six studies about ASBs. Way of beverage consumption was recorded by FFQ [19,41,58,59,60,61,62,63,64] and questionnaire [24]. We also extracted and compared the maximum and minimum intakes in studies. Methods of diagnosis and confirmed cases of CVDs were recorded, which included medical diagnosis [19,41,58,59,60,61,62,63,64] or self-reporting and medical diagnosis [24]. We counted adjustments in each study, nearly all studies assessed BMI (with the exception of [58,61]), smoking status (with the exception of [19]), energy intake (with the exception of [24,61]), and all studies assessed age, alcohol consumption, physical activity, and hypertension at baseline. Based on the Newcastle–Ottawa Scale, all studies were of a high quality (≥7 scores). Detail information is shown in Appendix A.

#### 3.2.1. SSBs and CVDs

A 17% higher risk of CVDs was shown in these cohorts (RR: 1.17; 95% CI: 1.12–1.23) with low evidence of heterogeneity (*I*^2^ = 14.7%, *P*_heterogeneity_ = 0.293) (Figure 2c). No particular study had a significant influence on the summary effect in a sensitivity analysis. The funnel plot was basically symmetrical (Figure 3b), and the Begg’s test and Egger’s test showed that there was no evidence of significant publication bias (Begg’s test: 0.0.049, Egger’s test: 0.069). 

Seven studies were included in the dose-response relationship analysis [58,59,60,61,62,63,64]. With each additional serving of SSB per day, the risk of developing CVDs increased by 9% (RR: 1.09, 95%CI: 1.07–1.12), with high moderate heterogeneity (*I*^2^ = 42.7%, *P*_heterogeneity_ = 0.065) (Figure 4c). No particular study had a significant influence on the summary effect in a sensitivity analysis. The publication bias was evaluated by the funnel plot (Figure 5b), the Begg’s test (*p* = 1.000), and the Egger’s test (*p* = 0.543). A linear relationship was observed between CVDs and SSBs intake (Figure 6c).

#### 3.2.2. ASBs and CVDs

As for ASBs intake, a 17% higher risk of CVDs was shown in these cohorts (RR: 1.17; 95% CI: 1.06–1.29), with moderate evidence of heterogeneity (*I*^2^ = 57.4%, *P*_heterogeneity_ = 0.021) (Figure 2d). Removing one large prospective cohort study [63], which accounted for all of the observed heterogeneity, the result changed to 1.20 (95% CI: 1.11, 1.29; *I*^2^ = 0.0%, *P*_heterogeneity_ = 0.463). We did not evaluate the publication bias, because of only eight groups of data in ASB intake and CVDs, less than 10 groups, according to the Cochrane Handbook. 

Four studies were included to analyze the dose-response relationship [24,58,61,63]. The summary RR for CVDs for each additional ASB serving per day was 1.08 (95%CI: 1.04–1.11), with moderate evidence of heterogeneity (*I*^2^ = 45.5%, *p* = 0.119) (Figure 4d). No particular study had a significant influence on the summary effect in a sensitivity analysis. We did not evaluate publication bias because there were less than 10 groups. There was a non-linear relationship between CVDs and ASBs intake (Figure 6d).

### 3.3. Beverage Intake and All-Cause Mortality

Eight prospective cohort studies with 999,689 participants from 3 countries were eligible for the analysis of beverage intake and the risk of all-cause mortality. There was one study for males [20], two studies for females [20,24], and six studies for both sexes [22,42,43,65,66,67]. All participants were adults, with ages ranging from 30 to 79 years old, and the follow-up durations were ranging from 6 to 34 years. We extracted eight groups of data from seven studies about SSBs and six groups of data from five studies about ASBs. The consumption of beverages was recorded by different ways, such as FFQ [20,22,43,65] and alternative ways [24,42,66,67]. The maximum and minimum intakes in studies were extracted and compared. Method of diagnosis and confirmed cases of all-cause mortality were recorded by medical diagnosis [20,22,24,42,43,65,66,67]. The adjustments in each study were counted. Nearly all studies assessed physical activity (with the exception of [24,66]), energy intake (with the exception of [24,43,66]), and fruits and vegetables (with the exception of [24,43,66]). All studies assessed age, BMI, alcohol consumption, and smoking status. Based on the Newcastle–Ottawa Scale, all studies were of a high quality (≥7 scores). Detail information is shown in Appendix A.

#### 3.3.1. SSBs and All-Cause Mortality

SSBs intake was associated with a 14% higher risk of all-cause mortality (RR: 1.14; 95% CI: 1.04–1.24), with high evidence of heterogeneity (*I*^2^ = 83.0%, *P*_heterogeneity_ < 0.001) (Figure 2e). We sequentially removed each study from the pooled analysis, but the heterogeneity remained stable. Because there were eight groups of data, less than 10 groups, we did not evaluate publication bias.

Four studies were included to analyze the dose-response relationship [22,65,66,67], with high evidence of heterogeneity (*I*^2^ = 86.3%, *p* < 0.001) (Figure 4e). After removing every study from the analysis, the heterogeneity remained stable. We did not evaluate publication bias because there were less than 10 groups. There was a non-linear relationship between all-cause mortality and SSBs intake (Figure 6e).

#### 3.3.2. ASBs and All-Cause Mortality

In these cohorts, ASBs intake was associated with a 15% higher risk of all-cause mortality (RR: 1.15; 95% CI: 1.06–1.24), with high evidence of heterogeneity (*I*^2^ = 78.9%, *P*_heterogeneity_ < 0.001) (Figure 2f). The heterogeneity remained stable after we sequentially removed each study from the pooled analysis. We did not evaluate publication bias because there were only six groups of data. 

Three studies were included in the dose-response analysis [24,66,67], with high evidence of heterogeneity (*I*^2^ = 76.9%, *p* = 0.013) (Figure 4f). After removing every study from the analysis, the heterogeneity remained stable. We did not evaluate publication bias because there were less than 10 groups. A non-linear relationship was observed between all-cause mortality and ASBs (Figure 6f).

### 3.4. Subgroup Analysis

According to the results, all three diseases showed heterogeneous changes in the subgroup analysis of adjustments, such as education, energy intake, fruit and vegetable intake, and baseline hypertension. In addition, in terms of T2D and CVDs, after the subgroup analysis of such factors as gender, follow-up duration, geographical region, number of participants, and BMI, there were heterogeneous changes in each group. These subgroup analyses may be the source of inter-study heterogeneity Appendix A.

## 4. Discussion

Findings from our systematic review and meta-analysis clearly showed that long-term consumption of SSBs and ASBs was positively associated with the risks of T2D, CVDs, and all-cause mortality. In the sensitivity analysis, the associations of SSBs consumption with T2D, CVDs, and all-cause mortality were stable, as well as ASBs consumption with all-cause mortality, while the relationships of ASBs consumption with T2D and CVDs were less evident. Publication bias was not discovered between SSBs consumption and T2D but was likely to exist between SSBs consumption and CVDs. Considering the health effects of SSBs and ASBs, the bias towards positive results seems reasonable, which showed that it was necessary to improve the quality of the evidence and give the rigorous interpretation of the results. For SBBs consumption, linear associations were found with risk of T2D and CVDs, and a non-linear association was found with risk of all-cause mortality, and their risks were increased by 27%, 19%, and 10% for each serving increase, respectively. For ASBs, non-linear associations with the risk all three outcomes and their risks were increased by 13%, 8%, and 7% for each serving increase, respectively.

Compared with the original research, we did the statistical analysis based on multiple studies, and the outcomes could provide more reliable guidance for nutrition practice. Although several meta-analyses have been performed, they have differences in some places. In recent years, there have been substantial studies about SSBs and a limited number of studies involved in ASBs, and we brought all the relevant studies (17 for T2D, 10 for CVDs, and 8 for all-cause mortality) into the present study. We focused on the prospective cohort studies in which more people would participate when comparing with the intervention studies. Furthermore, we extracted the data with the most adjustments in order to minimize the interference of other factors on the results, subgroup analyses were performed to provide multiple explanations for inter-study heterogeneity, and the dose-response relationships were analyzed to obtain accurate results of data. Compared with previous meta-analyses [25,27], six [39,47,50,51,55,57], three [39,47,55], four [19,60,62,64], and two [19,24] cohort studies were additionally included in our SSBs consumption-T2D, ASBs-T2D, SSBs-CVDs, and ASBs-CVDs meta-analyses, respectively. Although our findings for the positive associations of SSBs consumption with risk of T2D and CVDs were in correspondence to previous meta-analyses [3,25,26,27,28,45], we observed a non-linear dose-response association between ASBs consumption with risk of T2D, CVDs, and all-cause mortality, which was contradictory to a recent meta-analysis. The discrepancy might be explained by the previous meta-analysis only included limited studies and the studies about ASBs, T2D, CVDs, and all-cause mortality were infrequent.

The possible biological mechanisms involved in the relationships between SSBs consumption and risks of T2D, CVDs, and all-cause mortality might be explained by several reasons. First, the previous study showed that consumption of fructose, rich in SSBs, could increase de novo lipogenesis and visceral adipose deposition and induce more production of triglyceride, resulting in dyslipidemia, obesity, or metabolic syndromes [68]. Another explanation that was noteworthy was SSBs would contribute to incomplete compensation for energy at subsequent meals following the intake of liquid calories [69]. Additionally, SSBs consumption could increase the levels of blood glucose rapidly, and their high glycemic index is associated with the risk of T2D [70,71]. Furthermore, SSBs are related to an increase in blood pressure and uric acid, which contribute to the risk of cardiometabolic diseases [72,73]. Obesity, T2D, and increased cardio-metabolic may also contribute to the increased risk of CVDs and all-cause mortality associated with SSBs consumption. From another point, the blood glucose level, which is affected by SSBs, is associated with CV risk. This means SSBs not only affect cardiovascular disease through T2D but also cause cardiovascular disease through hyperglycemia [74,75].

The adverse effects of ASBs consumption on health outcomes are still unclear. It was noteworthy that ASBs intake was found to be not accompanied by typical and expected post-ingestive consequences, such as the release of insulin after meals, and therefore, this might degrade the capacity of sweet tastes to evoke these responses, resulting in higher blood glucose levels and more consequences, such as obesity, metabolic syndromes, and diabetes [23,76,77]. Furthermore, according to previous findings from animal studies, altered gut microbiota was found in mice by using artificial sweetener intervention, which was associated with the increased insulin resistance, and a similar phenomenon was also observed in humans [78]. However, the evidence is inadequate, and the conclusions still remain controversial [79,80,81]. Although the mechanism has not been fully established, our results indicated that reducing the consumption of ASBs would contribute to the health of humans, similar to the findings of SSBs.

There are several limitations to this study. Firstly, we extracted the effect sizes of the highest category versus the lowest category and achieved more intuitive risk outcomes, which might lead to larger risk outcomes because the results could not represent the basic consumption levels of beverages in the population. Secondly, the subgroups of dosages varied among different studies, which increased the difficulty of meta-analyses. Furthermore, dietary habits, especially for SSBs and ASBs intake, would change over time, resulting in uncertainties in their tracking process that would affect the final outcomes. Additionally, although the NOS scores in the studies were high, there was heterogeneity among them. Finally, some part of the meta-analyses was the lack of data support due to the limited number of studies. Our study also has some advantages over others. The meta-analysis was performed based on all prospective cohort studies published in recent years, which discussed the associations of SSBs and ASBs with T2D, CVDs, and all-cause mortality. The scope of our study was wide, the population was large, the quality of studies was at a high level, and the data with the most adjustments were selected for statistics so that our results were more comprehensive, representative, and reliable for clinical application.

## 5. Conclusions

Long-term consumption of SSBs and ASBs increased the risks of T2D, CVDs, and all-cause mortality in the present study. These findings of SSBs were stable in sensitivity analyses assessing the influence of population characteristics, potential residual confounding, and publication bias. On the contrary, although ASBs have also shown the ability to increase the risks of T2D, CVDs, and all-cause mortality, the quality of the evidence is limited by potential bias and heterogeneity due to the design of the study. More evidence is needed for further analysis and to improve the accuracy of the conclusions. Although the reasons for the connections were unclear, the results suggested that people should limit their consumption of SSBs and ASBs in order to reduce the adverse health effects.

## Figures and Tables

**Figure 1 nutrients-13-02636-f001:**
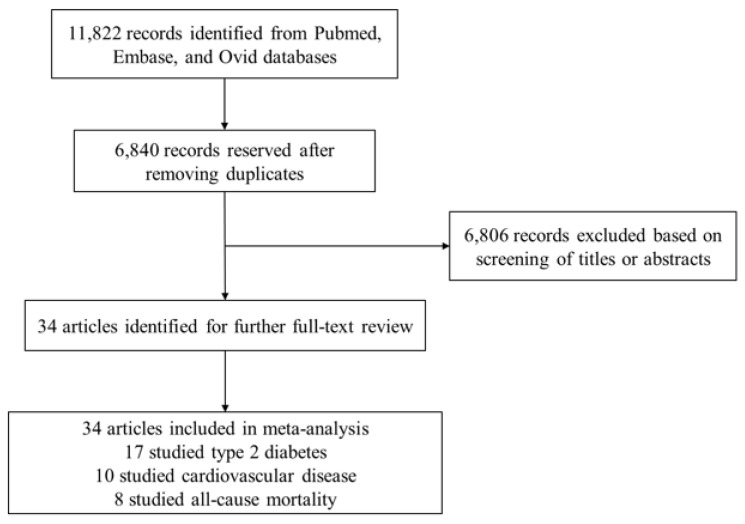
Selection of studies included in the meta-analysis.

**Figure 2 nutrients-13-02636-f002:**
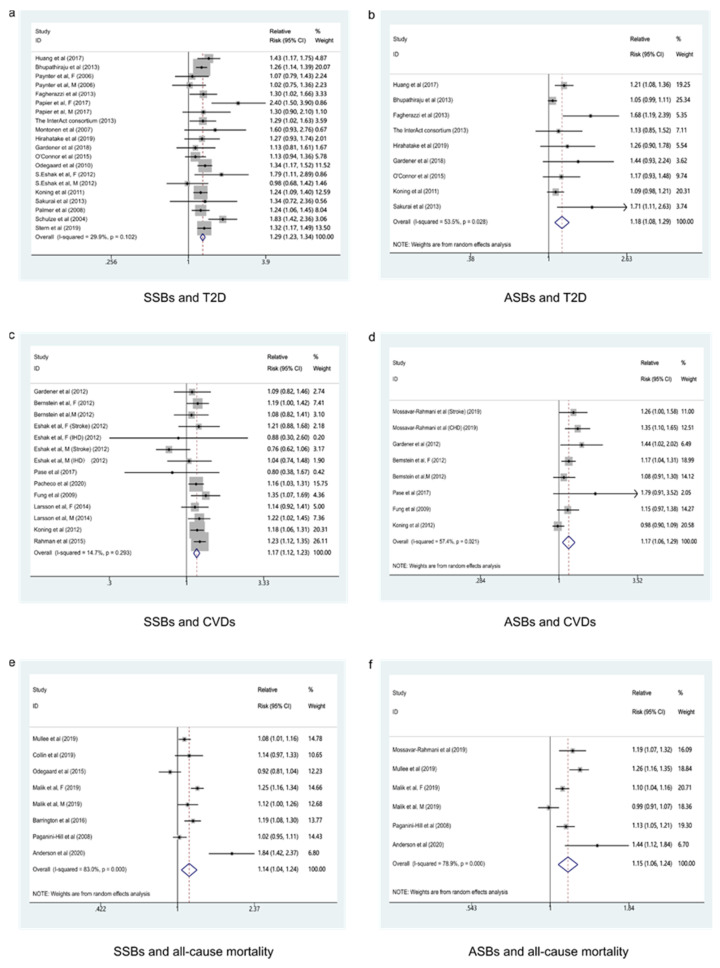
Forest plot of RR for the association of sugar-sweetened beverages and artificially sweetened beverages consumption with the risk of type 2 diabetes, cardiovascular diseases, and all-cause mortality. (**a**) Forest plot of RR for SSBs and T2D. (**b**) Forest plot of RR for ASBs and T2D. (**c**) Forest plot of RR for SSBs and CVDs. (**d**) Forest plot of RR for ASBs and CVDs. (**e**) Forest plot of RRfor SSBs and all-cause mortality. (**f**) Forest plot of RR for ASBs and all-cause mortality. SSBs, sugar-sweetened beverages; ASBs, artificially sweetened beverage; T2D, type 2 diabetes; CVDs, cardiovascular diseases.

**Figure 3 nutrients-13-02636-f003:**
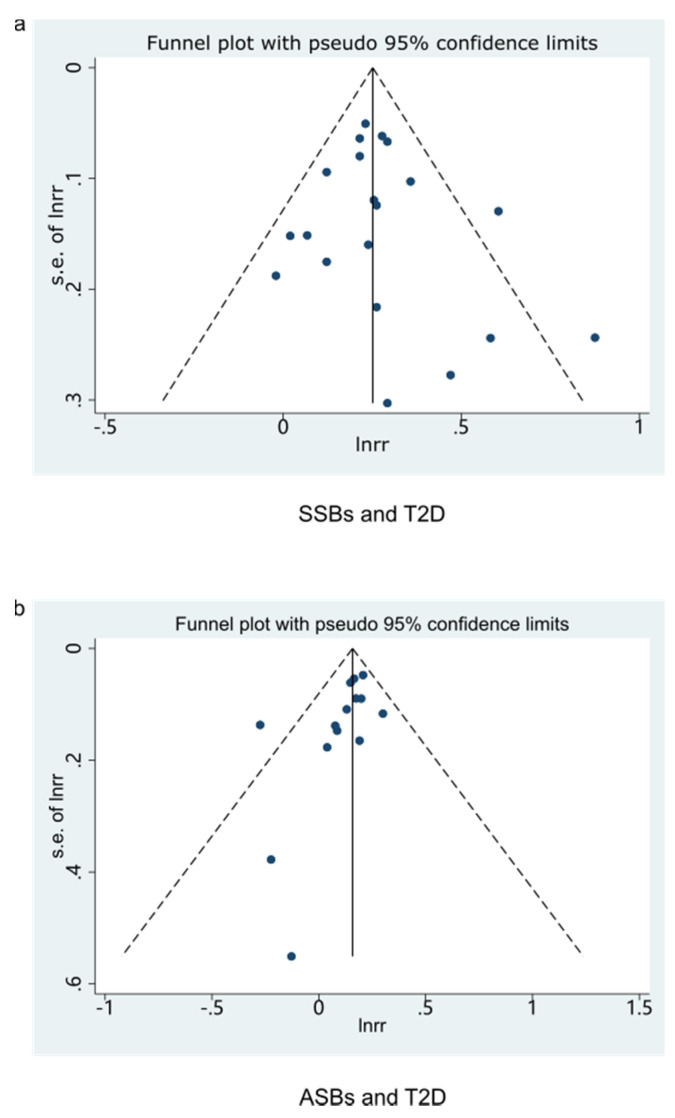
Funnel plot for examination of publication bias of studies (highest/lowest). (**a**) Funnel plot for SSBs and T2D. (**b**) Funnel plot for ASBs and T2D. SSBs, sugar-sweetened beverages; ASBs, artificially sweetened beverages; T2D, type 2 diabetes.

**Figure 4 nutrients-13-02636-f004:**
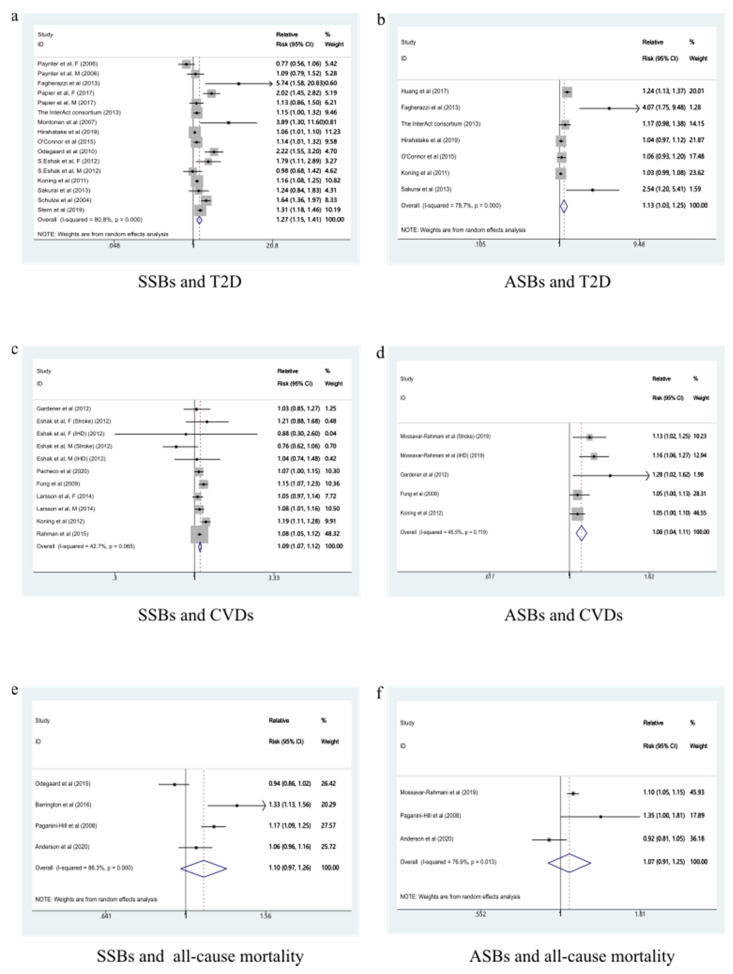
Forest plot of RR for each additional SSB and ASB serving per day and the risk of type 2 diabetes, cardiovascular diseases, and all-cause mortality. (**a**) Forest plot of RR for each additional SSB serving per day and the risk of T2D. (**b**) Forest plot of RR for each additional ASB serving per day and the risk of T2D. (**c**) Forest plot of RR for each additional SSB serving per day and the risk of CVDs. (**d**) Forest plot of RR for each additional ASB serving per day and the risk of CVDs. (**e**) Forest plot of RR for each additional SSB serving per day and the risk of all-cause mortality. (**f**) Forest plot of RR for each additional ASB serving per day and the risk of all-cause mortality. SSBs, sugar-sweetened beverages; ASBs, artificially sweetened beverages; T2D, type 2 diabetes; CVDs, cardiovascular diseases.

**Figure 5 nutrients-13-02636-f005:**
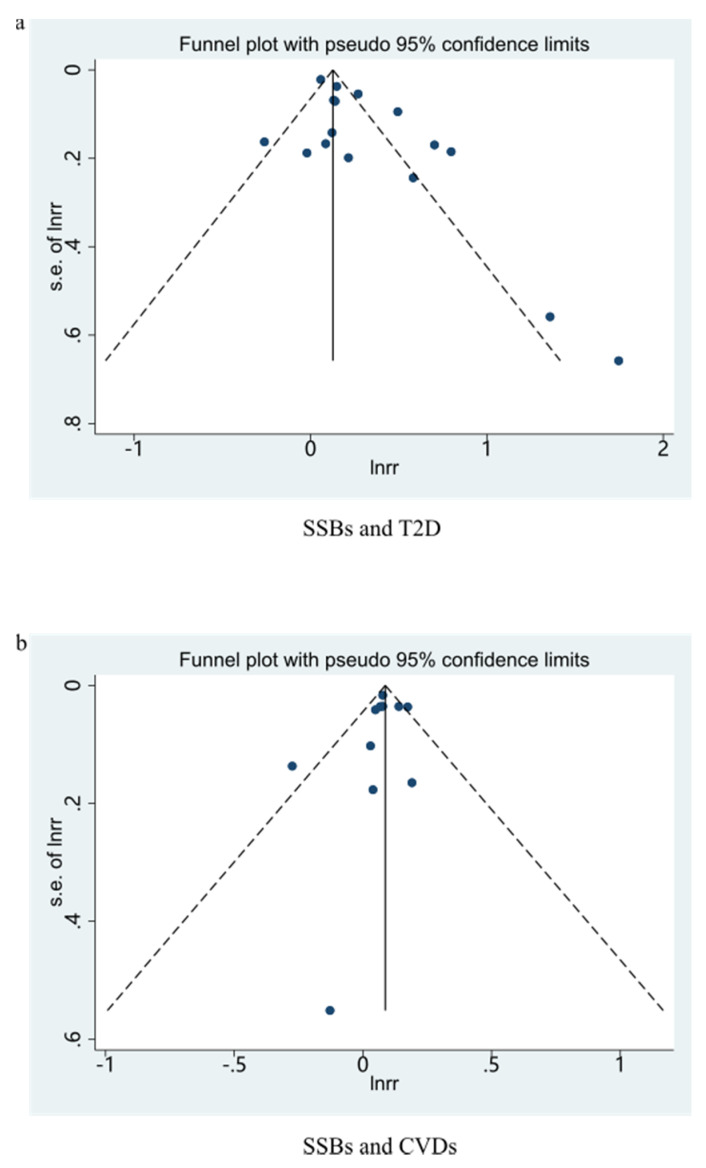
Funnel plot for examination of publication bias of studies (dose-response analysis). (**a**) Funnel plot for SSBs and T2D. (**b**) Funnel plot for SSBs and CVDs. SSBs, sugar-sweetened beverages; T2D, type 2 diabetes; CVDs, cardiovascular diseases.

**Figure 6 nutrients-13-02636-f006:**
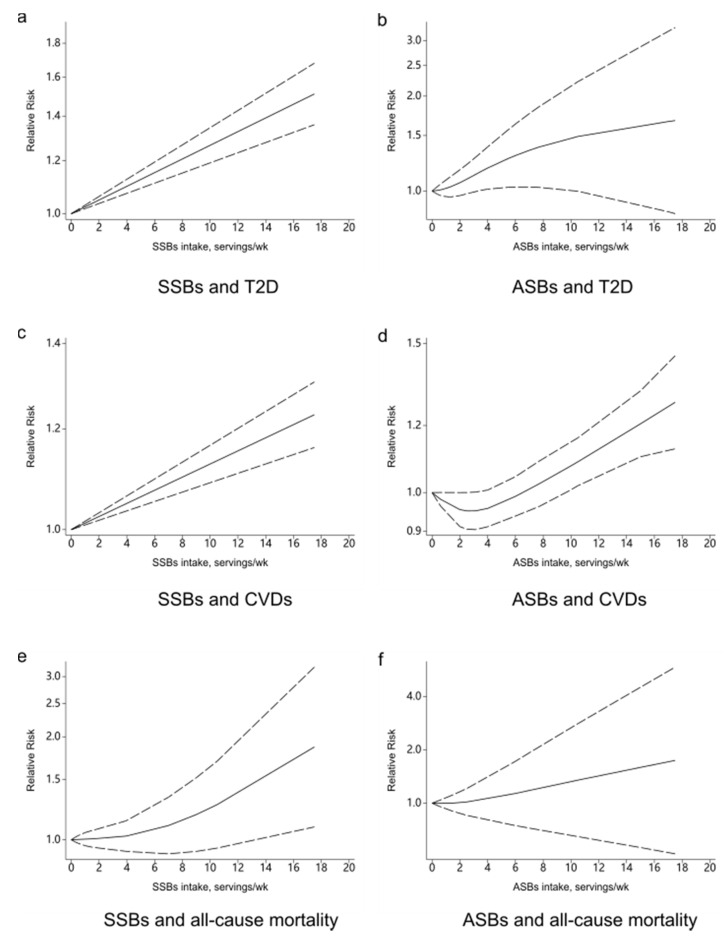
Dose-response relationship between sugar-sweetened beverages and artificially sweetened beverages consumption with the risk of type 2 diabetes, cardiovascular diseases, and all-cause mortality. (**a**) Dose-response relationship between SSBs and T2D. (**b**) Dose-response relationship between ASBs and T2D. (**c**) Dose-response relationship between SSBs and CVDs. (**d**) Dose-response relationship between ASBs and CVDs. (**e**) Dose-response relationship between SSBs and all-cause mortality. (**f**) Dose-response relationship between ASBs and all-cause mortality. SSBs, sugar-sweetened beverages; ASBs, artificially sweetened beverages; T2D, type 2 diabetes; CVDs, cardiovascular diseases.

## Data Availability

Not applicable.

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
