# Peer review of "Sugar- and Artificially Sweetened Beverages Consumption Linked to Type 2 Diabetes, Cardiovascular Diseases, and All-Cause Mortality: A Systematic Review and Dose-Response Meta-Analysis of Prospective Cohort Studies"

_nutrients, 2021, doi:10.3390/nu13082636_

Round 1

Reviewer 1 Report

The article on the consumption of sugary and artificially sweetened beverages related to type 2 diabetes, cardiovascular diseases and all-cause mortality, is of great interest due to the increase in consumption of these beverages in recent decades and their high prevalence with chronic diseases non-communicable. However, the article needs big changes:
-- Regarding the search strategy, we would find it interesting to use the Cochrane database to search for articles. Although the number of articles is high, very few were selected. So using the Cochrane database could contribute some more.
- It would be of interest to identify the search terms used, or if mesh terms were used for it; as well as the search formulas applied. We do not know if this data is in the supplementary information, as we have not had access to it.
- In the search strategy, how do the authors define relevant studies? Clarify that they are relevant studies.
- It would be important to measure the degree of agreement between the reviewers using the Kappa statistical calculation for each of the items on the selection sheet.
- In Figure 1, the reasons why 6806 articles were excluded should be indicated, given the significant number.
- It would seem important to us that a table be prepared explaining the PICO criteria.
- To analyze the biases, it would be recommended to apply Rob-2 by Cochranne (Risk of Bais Tool For Randomized Trials) to carry out a global evaluation of the different biases of the article.

Author Response

Dear reviewer,

Thank you for your advices. I would like to respond to your suggestions one by one.

  1. The reviewer’s comment:

The article on the consumption of sugary and artificially sweetened beverages related to type 2 diabetes, cardiovascular diseases and all-cause mortality, is of great interest due to the increase in consumption of these beverages in recent decades and their high prevalence with chronic diseases non-communicable. However, the article needs big changes:

-- Regarding the search strategy, we would find it interesting to use the Cochrane database to search for articles. Although the number of articles is high, very few were selected. So using the Cochrane database could contribute some more.

The author’s answer:

We use PubMed, Ovid, and Embase database to search the relevant studies because they are authoritative and have a large number of researches. Meanwhile, we have referred to some articles for search strategies, which also chose these databases as data resource (such as: Consumption of sugar sweetened beverages, artificially sweetened beverages, and fruit juice and incidence of type 2 diabetes: systematic review, meta-analysis, and estimation of population attributable fraction, doi: 10.1136/bmj.h3576; Television Viewing and Risk of Type 2 Diabetes, Cardiovascular Disease, and All-Cause Mortality A Meta-analysis, doi: 10.1001/jama.2011.812; Fish consumption and the risk of cardiovascular disease and mortality in patients with type 2 diabetes: a dose-response meta-analysis of prospective cohort studies, doi: 10.1080/10408398.2020.1764486. Epub 2020 May 15).

  1. The reviewer’s comment:

- It would be of interest to identify the search terms used, or if mesh terms were used for it; as well as the search formulas applied. We do not know if this data is in the supplementary information, as we have not had access to it.

The author’s answer:

Search terms are in the supplementary materials: ("soda" OR "pop" OR "juices" OR "juice" OR "drink" OR "drinks" OR "beverage" OR "beverages") AND ("diabetes" OR "coronary heart disease" OR "CHD" OR "angina" OR "heart disease" OR "ischemic heart disease" OR "ischaemic heart disease" OR "IHD" OR "myocardial ischemia" OR "myocardial infarction" OR "MI" or "coronary artery disease" OR "atherosclerosis" OR "cardiovascular disease" OR "CVD" OR "vascular disease" OR "vascular event" OR "stroke" OR "ischemic stroke" OR "cerebral infarction" OR "cerebrovascular disease" OR "all-cause mortality" OR "death" OR "mortality" OR "mortalities" OR "fatal" OR "total mortality" OR "survival") AND ("prospective" OR "prospectively" OR "cohort" OR "cohorts" OR "longitudinal" OR "observational" OR "observation" OR "Follow-up" OR "case-cohort" OR "nested case-control". (Page 14, line 451-459)

  1. The reviewer’s comment:

- In the search strategy, how do the authors define relevant studies? Clarify that they are relevant studies.

The author’s answer:

In the search strategy, we first used search terms for screening, then read the abstract to determine the content of the article, and finally read the full text for further verification.(Page 3, line 103-147)

  1. The reviewer’s comment:

- It would be important to measure the degree of agreement between the reviewers using the Kappa statistical calculation for each of the items on the selection sheet.

The author’s answer:

Reviewers followed consistent screening rules and screened the articles strictly as described in the method.(Page 3, 115-147)

  1. The reviewer’s comment:

- In Figure 1, the reasons why 6806 articles were excluded should be indicated, given the significant number.

The author’s answer:

In Figure 1, 6806 articles were excluded, because their articles or abstracts did not meet the standards. (Page 3, 115-147). And they were irrelevant to this study.

  1. The reviewer’s comment:

It would seem important to us that a table be prepared explaining the PICO criteria.

The author’s answer:

We would like to explain the PICO criteria for you: for the whole study, P is "healthy people", I is "intake of SSBs or ASBs", C is "no/little intake of SSBs or ASBs", O is "T2D, CVDs, and all-cause mortality". And the specific content is presented in the file called “Tables”. We extracted every study’s information and make them in the table.

  1. The reviewer’s comment:

To analyze the biases, it would be recommended to apply Rob-2 by Cochranne (Risk of Bais Tool For Randomized Trials) to carry out a global evaluation of the different biases of the article.

The author’s answer:

We learned about the tool (ROB 2) you recommended, but we found that this tool was for RCT(randomized control trials). Our study focused on prospective cohort studies, and ROB 2 don't apply to them. Newcastle-Ottawa quality assessment scale (NOS) is appropriate for observational studies. We measured the quality of studies by NOS and results were presented in the file, called "Tables". Because space is limited, we put the data into information tables. The last column of table 1, 2, and 3 is "NOS scale", which analyzed the bias. If a detailed scoring form is required, we will provide it.

Thank you again for your advices and we are honored to have your guidance. If you have any other advices, look forward to our communication.

Best regards,

Yantong Meng.

Reviewer 2 Report

Yantong Meng et al. performed a meta-analysis on 34 studies. The findings indicate that increased consumption of sugar-sweetened beverages (with linear relationships) and artificially sweetened beverages (with non-linear relationships) is associated with risk of T2D and CVDs. Moreover, increased consumption of SSBs and ASBs is associated, with non-linear relationship, with all-cause mortality.

The study is interesting, and quite original. The analysis was well done.

This reviewer raises few issues that the authors have to address.

1- The relationship between sugar-sweetened beverages consumption and CVDs may not necessarily pass through the development of T2D, but also through simple hyperglycemia. Indeed, it was observed that tight glycemic control improves the CV outcome after ACS even in non-diabetic hyperglycemic subjects (Journal of Clinical Endocrinology and Metabolism Volume 97, Issue 3, March 2012, 933-942. Doi: 10.1210 / jc. 2011- 2037 - Journal of Diabetes Research, 2018, art. No. 3106056. doi: 10.1155 / 2018/3106056). This important issue, and above references should be added and commented on in the discussion.

2- The authors pointed out the important limitations of the study. Therefore, the conclusions are cautious. Nonetheless, the authors should emphasize the limitations more even in the conclusions

3- A linguistic revision of the text by a native English speaker is suggested.

Author Response

Dear editor,

Thank you for the issues that you pointed out. I read your suggestions carefully and I added them to my study. It's a great honor to get your advices.

  1. The reviewer’s comment:

Yantong Meng et al. performed a meta-analysis on 34 studies. The findings indicate that increased consumption of sugar-sweetened beverages (with linear relationships) and artificially sweetened beverages (with non-linear relationships) is associated with risk of T2D and CVDs. Moreover, increased consumption of SSBs and ASBs is associated, with non-linear relationship, with all-cause mortality.

The study is interesting, and quite original. The analysis was well done.

This reviewer raises few issues that the authors have to address.

- The relationship between sugar-sweetened beverages consumption and CVDs may not necessarily pass through the development of T2D, but also through simple hyperglycemia. Indeed, it was observed that tight glycemic control improves the CV outcome after ACS even in non-diabetic hyperglycemic subjects (Journal of Clinical Endocrinology and Metabolism Volume 97, Issue 3, March 2012, 933-942. Doi: 10.1210 / jc. 2011- 2037 - Journal of Diabetes Research, 2018, art. No. 3106056. doi: 10.1155 / 2018/3106056). This important issue, and above references should be added and commented on in the discussion.

The author’s answer:

Thank you for your affirmation about our work. We are so glad to have the chance to communicate with you.

In page 13, line 406-409, I added the sentences “From another point, the blood glucose level, which is affected by SSBs, is associated with CV risk. This means SSBs not only affects cardiovascular disease through T2D, but also causes cardiovascular disease through hyperglycemia[76, 77].”

  1. The reviewer’s comment:

- The authors pointed out the important limitations of the study. Therefore, the conclusions are cautious. Nonetheless, the authors should emphasize the limitations more even in the conclusions

The author’s answer:

In the conclusion of page 14, line 444-445, I added “More evidences is needed for the further analysis and to improve the accuracy of the conclusions.” to supplement the content.

  1. The reviewer’s comment:

- A linguistic revision of the text by a native English speaker is suggested.

The author’s answer:

Due to limited time for modification, we don’t finish the modification of the language of the full text. And we will continue to do it.

Best regards,

Yantong Meng.

Round 2

Reviewer 1 Report

The authors have made the appropriate modifications based on the reviewers´ comments. Therefore, the article is ready to be published. 

Reviewer 2 Report

The authors addressed all issues raised by this reviewer.